# On the Other Side of the Looking Glass: COVID-19 Care in Immigration Detention

**Dora Schriro**

Independent Researcher, New York, NY 10464, USA; dora.schriro@gmail.com

**Abstract:** Immigration Detention is a patchwork of public and private correctional facilities overseen by ICE, a federal enforcement agency. In June 2021, ICE detained 16,460 adults in 121 facilities in 38 states, frequently alongside pretrial and sentenced inmates and U.S. Marshals Service prisoners, under varying conditions ICE established with five different sets of detention standards, all of them based on corrections case law and in effect today. Detainees have not fared well in ICE's custody, especially during the pandemic. In CY2020, ICE processed 137,749 detainees, tested only 80,200 for COVID-19 (58%), and recorded 8622 positive cases (11%) at over 100 facilities. Most testing positive for COVID-19—7687 (89%)—contracted the virus in ICE custody, including eight detainees who died. An additional 14,728 detainees (18%) had one or more conditions placing them at high risk for severe illness due to COVID-19 of which ICE only released 5801 (39%). This paper utilizes ICE data and documents on government websites to evaluate ICE's approach to detention management and explore its impact on conditions of detention and how it impeded its readiness and response to the pandemic. It concludes with recommendations that ICE decrease reliance on detention and decriminalize its policies and practices.

**Keywords:** crimmigration; incarcergration; decriminalization; detention; detention standards; alternatives to detention; conditions of detention; coronavirus





## 1. Prologue

*The Body Ritual among the Nacirema*

The Body Ritual Among the Nacirema ([Miner 1956](#)) is an anthropological essay, a culture-free description of a man, a Nacirema, which is American spelled backwards. He is standing at the bathroom sink, looking at himself in the mirror while shaving. An observer from another time and place mistakes the mirrored medicine cabinet for a magical box in which many charms and potions are kept, and before which the man chants and sways every morning.

Before joining the U.S. Department of Homeland Security (DHS) to serve as Senior Advisor to DHS Secretary Napolitano on Detention and Removal, I was Warden of a city jail, then Commissioner of two city and two state correctional systems. As I toured the first of many immigration detention facilities operated on behalf of Immigration and Customs Enforcement (ICE), I walked through it as would a newly admitted detainee become acquainted with her place of confinement, starting at the sallyport, then Intake, through the medical area, to the housing units, a combination of 50-bed cellblocks and 100 bunks dormitories in which people were packed, nose to toes, under conditions as severe and secure as high-custody correctional facilities, locked-in as many as 23 h every day. I moved through segregation housing, still operating as it had when it was a prison, then food services, the laundry, law library, visitation, and commissary, onto the recreation yard. This detention center looked like a correctional facility in almost every respect and, initially, appeared to operate as one would ([Goffman 1961](#)) and, make no mistake, that would not be a good thing. Detainees surrendered all their personal property at admission along with any semblance of their personal identities; in exchange, each was issued a

photo ID with an "Alien Number" by which they would now be known. After a cursory assessment of their immigration and criminal history (when there was any) they exchanged their street clothes for a couple of jumpsuits in one of several colors associated with their custody classification—in my opinion, ill-advised in any settings—and the first departure from correctional practices that I noted that day—blue uniforms for low security, orange uniforms for medium security, and red uniforms for maximum security—and then were assigned to housing units with others in same custody classification.

On the recreation yard, the differences were more pronounced. A large group of men, all of them wearing red unforms, supposedly to warn others of their violent nature, were supervised by only one detention officer. Some of them were playing a game of horseshoes, tossing real horseshoes around sharpened metal stakes that had been pounded into the ground, as others stood around them and watched. A correctional facility would never release any number of truly dangerous inmates together onto a recreation yard, certainly not a yard enclosed in just a six-foot chain link fence and the one officer in the area, and under no circumstances with heavy metal objects at hand that could—and likely would—be used as weapons if they were violent. When it was time to go back indoors, there was no inventory of the equipment or search of the detainees for contraband. The officer simply escorted them back to their housing unit, through the main corridor where there were a lot of detainees in blue and orange jumpsuits—without incident. The people who ICE has continually characterized as the "worst of the worst," and not trustworthy of being assigned to open housing, and certainly not community supervision under any circumstances, had played by the rules, literally. In fact, most detainees always do.

The vast majority of the people in ICE's custody are contributing members of intact, extended families, with job skills, employment histories, and community ties. They do not want any trouble, only the opportunity to be heard and hopefully, secure relief. Unfortunately, ICE policy and practice is a self-fulfilling prophesy. When the government locks up people who are pursuing civil remedies through the immigration court in jails and prisons, dressed in jumpsuits with their movement monitored moment to moment by uniformed guards, or assigns them to community supervision with an electronic monitoring device tethered to their ankle, we conclude they must be criminals. Why else would the government treat them as such? If ICE were to house migrants on college campuses or in hostels, rectories, training centers and worksites, and similar settings, dressed in street clothes, and had them check in periodically with a coach or an advisor, should any of these provisions actually be warranted, our opinion of them would be as different as the treatment they receive. ICE has never assessed risk correctly or responded proportionately, and despite its unfounded exaggerations as to detainees' dangerousness, many had never been convicted of a crime before they were detained, cause no trouble during their detention, and do not engage in criminal activities of any kind after their release (Schriro 2009).

ICE's oversight and operation of immigration detention is reminiscent of the narrator who misinterpreted the Nacirema's moves and motives as he went about his morning rituals. ICE goes through many of the motions associated with criminal incarceration without an apparent understanding or appreciation of the substantive differences between detainees and inmates, and immigration enforcement's distinctly different role and responsibilities for *civilly* detained individuals in its custody—especially those at heightened risk of serious illness, life-altering complications, and death from COVID-19 during the pandemic. ICE's criminalization of the immigrant deprives all the people in its custody of their rights under international and federal law and absolves Immigration Enforcement of its responsibilities to the detained (Bowling and Westenra 2018).

## 2. Introduction

Immigration Detention is a patchwork of public and private correctional facilities overseen by ICE, a federal enforcement agency. In June 2021, ICE detained 16,460 adults in 121 facilities in 38 states, frequently alongside pretrial and sentenced inmates and U.S. Marshals Service prisoners, under conditions ICE established in five different sets of

detention standards it uses and which are based on corrections case law. Detainees do not fare well in ICE's custody, particularly during the pandemic. In calendar year (CY) 2020, ICE processed 137,749 detainees, tested only 80,200 for COVID-19 (58%), and recorded 8622 positive cases (approximately 11%) at over 100 facilities nationwide, including eight detainees who died from the coronavirus while in ICE custody. Among those who tested positive for COVID-19, 7687 (89%) were exposed to the virus while in ICE custody. An additional 14,728 detainees (18%) had one or more conditions that placed them at high risk for severe illness due to COVID-19 of which, ICE released only 5801 high-risk individuals (39%) (GAO-21-414).

I believe ICE can do better. As a matter of law, it must. Immigration detainees are held pending *civil* proceedings in the immigration court. Their detention must not be punitive and their access to healthcare must meet the community standard. ICE has not met either of these requirements.

We are at a flex point. There is need to act, and there is opportunity to do so. One, the pandemic continues to threaten the public's health, particularly in areas of the country where vaccination rates remain low. ICE also reported low rates among detainees nationwide, regardless of the facilities' locations (Melugin 2021). This is not surprising given the state of its healthcare system. Two, the size of the detained population remains relatively low, but it has begun to rise again. Now is the time to eliminate as many beds as possible and change the nature of those that remain. Three, the new Administration is open to the idea of creating a civil, civil system of immigration enforcement.

This paper considers several of the ways in which ICE has misapplied corrections case law, policy and practices to the detriment of the detainees in its custody, particularly during the pandemic. It utilizes ICE data and documents on government websites to consider ICE's approach to detention management, how it has impacted conditions of detention, and impeded its readiness and response to the pandemic. It is recommended that reliance on Immigration Detention is decreased, that it is decriminalized, and ICE is held accountable for its activities and outcomes. In short, this is a brief look in that mirror, at ICE, the aggregated impact of crimmigration law on a quasi-punitive system of immcarceration, and us.

## 3. COVID-19 and the CDC's Standard of Care for Correctional and Detention Facilities

The U.S. Centers for Disease Control and Prevention (CDC) is the nation's health protection agency (CDC 2019). CDC Guidance is the community standard of healthcare for treating coronavirus, and the CDC Interim Guidance on Management of Coronavirus Disease 2019 (COVID-19) in Correctional and Detention Facilities (CDC Guidance) is the community standard for corrections and immigration detention (CDC 2020). Every detainee in ICE's custody has a right to receive this level of care.

On 11 March 2020, the World Health Organization declared the novel coronavirus (COVID-19) outbreak a global pandemic. On 23 March 2020, the CDC issued its initial CDC Guidance. The CDC has continued to tailor public health responses to coronavirus for incarcerated populations throughout the course of the pandemic.

Chief among its recommendations to prevent the spread of COVID-19 in incarceral settings, the CDC urged correctional and detention facilities to practice extreme social distancing, continual and correct use of personal protective equipment (PPE), and heightened sanitation and vigorous hygiene facility-wide, coupled with cohorting and screening for symptomatic individuals, testing of asymptomatic individuals, quarantine and contact tracing, and when it became available, vaccination. The CDC has never proposed however, that correctional and detention facilities release at-risk persons in their custody, deferring to the executive, legislative and judicial branches of city or county, state, and federal government to make those decisions. Some have been critical of its silence: Instead of centering on public health, it appeared to them that the CDC was preoccupied with the impact of such a recommendation on traditional enforcement priorities (Harvard Law Review 2021).

## 4. Civil Detention v. Criminal Incarceration

### 4.1. The Applicable Legal Standard for Immigration Detainees

Convicted prisoners are protected by the Eighth Amendment to the U.S. Constitution, and application to the states by the Fourteenth Amendment, which prohibits the infliction of cruel and unusual punishments on those persons. To establish a violation of the Eighth Amendment, a prisoner must show both a deprivation of a basic human need (Helling v. McKinney 1993) and deliberate indifference (Wilson v. Seiter 1991). In the context of medical or mental health care, she must demonstrate "deliberate indifference to serious medical needs (Estelle v. Gamble 1976).

Pretrial prisoners are protected by the Due Process Clauses of the Fifth and Fourteenth Amendments against any conditions that constitute "punishment (Bell v. Wolfish 1979)". They are afforded at least as much protection as are sentenced inmates regarding medical care (City of Revere v. Massachusetts General Hospital 1983). Deliberate indifference of correctional officials to the serious medical needs of a pretrial prisoner is a violation of her due process.

Immigration detainees are civil detainees held pursuant to civil immigration laws. Their protections are also derived from the Fifth Amendment, shielding persons in the custody of the United States from conditions that amount to punishment (Wong Wing v. United States 1896). ICE may detain non-citizens during the removal process (Fong Yue Ting v. United States 1893) but, because immigration detention is not punishment, its detention must not be excessive in relation to ICE's noncriminal purposes (Zadvydas v. Davis 2001). To do so is improperly punitive, thus unconstitutional (United States v. Solano 1987). Immigration detainees must be afforded the same (Edwards v. Johnson 2000) or superior (Jones v. Blanas 2004, Youngberg v. Romeo 1982) level of protection as are pre-trial prisoners.

### 4.2. Crimmigration Law and the Quasi-Punitive System of Immaceration

Our blurring of criminal enforcement and immigration control has given rise to a system of crimmigration law. Similarly, our treatment of civilly detained people in immigration proceedings as if they are criminally charged, using criminal incarceration for the purpose of immigration detention, serves to validate public perception and fortify public acceptance of excessive immigration practices, giving way to the quasi-punitive system of immcarceration, with which we grapple today: a random collection of correctional practices with many of its punitive characteristics, unsubstantiated beliefs about deterrence, and none of its due process protections. Preventive justice is neither preventive nor just (Cole 2014). Many detainees seek asylum. Others seek reunification with family members who preceded them. In most cases, migrants are certain that they have no choice but to immigrate, the conditions in the counties of their origin are so detrimental that they are compelled to make the harrowing trip to our borders and surrender themselves thus initiate the process of lawful entry. Their fear for survival has also become ours. Xenophobia informs our policies and procedures and over time, immigration detention has become a deprivation as severe as removal itself (Kalhan 2010).

Immigration enforcement also lacks the criminal justice system's checks and balances, measured practices upon which the disenfranchised depend. Whereas there is considerable discretion distributed across decision-makers in the criminal justice system from the arresting officer, prosecuting attorney, bail bondsman, pre-trial services, and arraignment and trial courts at the front-end to pre-release services, victim advocates, and parole board at the back-end, detain or release decision-making is concentrated primarily within DHS, and controlled largely by Customs and Border Patrol (CBP) and ICE; the focus of their activities along the northern and southern borders, and the interior, respectively. CBP and ICE also regulate the conditions of confinement in CBP patrol stations and ICE detention facilities and operate their respective holding and detention facilities. As CBP and ICE, both immigration enforcement agencies, prefer removal to relief, and detention is an expedited means to that end, one that only they control, they exercise control with impunity. Only the

U.S. Citizenship and Immigration Services (USCIS) within DHS, its charge to adjudicate non-citizens requests for immigration benefits, or the Executive Office for Immigration Review (EOIR) within the U.S. Department of Justice (DOJ), the nation's immigration court system, can change the outcome of immigration cases. Given the disparity in the size, staffing, and status of each of these agencies, it does not happen as often as it should.

In many cases, detention is also mandatory as a matter of law. Mandatory Detention refers to provisions of the Immigration and Nationality Act (INA), § 236(c) and § 235(b), which state non-citizens with certain criminal convictions are not entitled to a bond hearing, they must be detained by ICE, and shall remain detained while removal proceedings are pending against them. As indicated in Table 1, on 14 June 2021, there were 16,460 people in ICE's custody of which, 11,570 (70%) were mandatorily detained although 9510 (58%) had no criminal convictions.[1] ICE refers to this group as "No ICE Threat Level (ICE 2021a). These changes to the INA were made in the 1990's, the same period of time in which Truth-in-Sentencing, Three Strikes, and Juvenile Justice "reforms" were enacted, many of those provisions since reversed. I think it would be worthwhile to reconsider the validity of the assumptions that brought about these amendments as well as beginning with a review of the 28% of detainees who had no convictions and were mandatorily detained nonetheless.[2]

**Table 1.** Average Daily Threat Level (ICE 2021b).

| Facility | ICE Threat Level[3] | | | | ADP Total, all Threat Levels[4] | Mandatory Detention |
| --- | --- | --- | --- | --- | --- | --- |
| | Level 1 High Risk | Level 2 Medium Risk | Level 3 Low Risk | No ICE Threat Level No Risk | | |
| TOTALS | 4138 (25%) | 1460 (9%) | 1352 (8%) | 9510 (58%) | 16,460 (100%) | 11,570 (70%) |

It is clear that ICE has taken other measures to keep people detained. One notable example is ICE's revision of its Custody Classification detention standard. As developed by the Immigration and Naturalization Services (INS), it was an objective process, an assessment based on facts—whereby an opinion, even informed opinion (based on profiling, familiarity, personal experience, etc.) is different from fact, therefore irrelevant for detainee classification (ICE 2002b)—to a subjective process where "discernable" facts—such as nothing more than a tattoo to establish gang membership—as are acceptable (ICE 2019b). ICE also repeatedly adjusted its risk assessment instrument's algorithm, continually modifying it to raise the custody scores of as many detainees as possible to avoid releasing them (Koulish 2016). When that still did not eliminate as many detainees as ICE believed should remain in custody, ICE revised its already limited range of recommendations in 2018, striking all but one outcome regardless of the detainees' risk score: Detain (Rosenberg and Levinson 2018).

In FFY2019, pre-pandemic (ICE 2019d), ICE's average daily detained population (ADP) reached 50,165 detainees. In FFY2020, mid-pandemic (ICE 2020a), after enforcement activities had been scaled back considerably, the ADP dropped to 19,068 detainees (<62%). ICE's ADP began to rebound and by June 2021, rose to 26,222 detainees (>28%) by mid-June in FFY2021 (ICE 2021a), with the expectation its ADP would continue to increase unless there was a marked change in enforcement policy.

In fact, the pandemic brought about a significant shift in both federal policy and state and local practice. A change in federal public health policy altered CBP's apprehensions of migrants. Another in the courts and correctional systems impacted ICE's arrests. Together, they account for most of the precipitous drop in "book-ins," the combined annual totals of CBP apprehensions and ICE arrests, from 510,854 migrants in FFY 2019 to 182,869 in FFY2020 (<36%) (U.S. ICE 2021).

At the federal level of government, the prior Administration utilized sections 362 and 365 of the Public Health Service Act, 42 U.S.C. §§ 265 and 268, to suspend "the introduction

of persons into the United States" beginning in March 2020, purportedly to prevent the introduction of COVID-19 into the country. Named the Migrant Protection Protocols (MPP) but better known as the "Remain in Mexico Policy," its impact was immediate. Pursuant to MPP, most migrants along the southwest border, many of whom sought asylum, must remain in Mexico, currently a year or more, until such time as they are called to appear in immigration court. In August 2021, over 70,000 people seeking asylum were waiting for a date to be heard (Morrissey 2021). Immediately upon taking office in January 2021, the current Administration reversed the MPP. The states of Texas and Missouri sued, seeking its reinstatement, and they prevailed in the federal district court. The Administration petitioned the U.S. Supreme Court to grant a stay, and in August 2021, the Court denied its application (Biden et al. v. Texas et al. 2021) thereby keeping in place MPP for now.

At the state and local levels of government, both the courts were clearing their confined criminal dockets and correctional systems were reducing their jail and prison populations, especially of medical vulnerable individuals, through various release mechanisms. With fewer inmates remanded to correctional facilities, there were also fewer individuals to turn over to ICE and a number of them both pre-trial or pre-plea and sentenced were medically vulnerable. Transferring at-risk individuals, especially those who had neither plead nor proven guilty, from one authority to another was contra-indicated by the Court and the CDC. Nevertheless, ICE continued to take them into their custody upon their release from the criminal justice system although many of them would be released to its ATD program.

It made a measurable difference. In FY 2019, ICE monitored 83,186 adult ATD participants (ICE 2019d); in FY2020, 85,415 adults (>3%) (ICE 2020a); and in FY2021 TD, 103,933 adults (>18%) (ICE 2021a).

Notwithstanding the appreciable decrease in book-ins through the course of the pandemic, COVID-19 more than doubled the time that migrants remain in its custody and under its supervision. Pre-pandemic, delays in the immigration court's detained and non-detained dockets were considerable. Coronavirus compounded both backlogs. The average length of stay (ALOS) in detention rose from 34 days in FY2019 (ICE 2019d), to 63 days in FY2020 (>54%) (ICE 2020a), to 60 days in FY2021 TD (<5%) (2021a). The average length of time in an ATD program (ALIP), was far worse, rising from 352 days in FY2019, (ICE 2019d) to 816 days in FY2020 (>57%) (ICE 2020a), to 788 days in FY2021 TD (<4%) (ICE 2021a).

## 5. The Immcarceration of Immigration Detention

The shift in immigration policy from "Catch and Release" to "Catch and Remove" in 2005, left ICE scrambling for additional beds to detain the burgeoning non-criminal population. At the ready were thousands of public and private prison beds that had been built the decade before to accommodate the growth in the inmate population brought about by state and federal sentencing initiatives in the 1990's.

ICE acquired many of these beds and did so without the benefit of population forecasting, multi-year capital construction plans, a scope of work with clear selection criteria and agency-specific operating assumptions, or competitive bidding, all of which are widely recognized management tools. Instead, it did what was expedient to meet its mandate: it got those beds by various means to deter and detain. In 2009, when I conducted the nationwide review of immigration detention at the direction of DHS Secretary Napolitano (Schriro 2009), ICE had secured space in over 300 jails and prisons to house as many as 31,000 adults daily, facilities still staffed by correctional personnel, and operating as correctional facilities with all its policies and procedures—counts, controlled movement, searches, shakedowns, and the like—intact. To this, ICE only added the requirement that all facilities housing an average of ten or more detainees would also comply with its detention standards.

*ACA Adult Local Detention Correctional Standards v. ICE Immigration Detention Standards*

In September 2000, the Immigration and Naturalization Services (INS), consisting of USCIS, CBP and ICE, each of which would become stand-alone agencies within DHS

in 2002, promulgated the first detention standards for facilities housing immigration detainees. The INS selected the American Correctional Association (ACA) standards for adult local detention facilities (ALDF), based upon corrections case law for pretrial and locally sentenced prisoners,[5] as the prototype for its 2000 National Detention Standards (NDS) (ACA 2004, 2016). The INS intended detention standards to establish consistent conditions of confinement, program operations, and management expectations within its detention system. Although that was INS' intent, the 2000 NDS made allowances for non-dedicated facilities, merely encouraging them to consider those procedures useful as guidelines, (ICE 2002a). When ICE was formed in 2002, the agency continued to operate immigration detention utilizing the 2000 NDS.

In 2004, the ACA transitioned to performance-based detention standards, and in 2008 ICE published the first of several sets of performance-based National Detention Standards (PBNDS), replicating those of the ACA. As had the ACA, ICE incorporated expected outcomes for each standard and expected practices required to achieve them so as "to improve safety, security, and conditions of confinement for detainees (ICE 2008a)".

In 2011, ICE revised the 2008 performance-based detention standards, incorporating changes made following the release of the 2009 Schriro Report (Schriro 2009) and to address outstanding recommendations. ICE said of the 2011 PBNDS standards, "It represents an important step in detention reform (ICE 2011a)".

In 2016, ICE revised the 2011 detention standards "to ensure consistency with federal legal and regulatory requirements as well as prior ICE policies and policy statements," incorporating provisions of the Prisoner Rape Elimination Act (PREA) and Section 504 of the Rehabilitation Act prohibiting discrimination on the basis of disability, as well as changes to the operation of Special Management Units, expansion of language services, and other ICE and ERO Directives, Memoranda and Policy Statements (ICE 2016a).

In 2019, ICE issued National Detention Standards (NDS) for Non-Dedicated Facilities. Non-dedicated facilities house one or more other populations typically, inmates, often from more than one jurisdiction, occasionally military prisoners, and increasingly U.S. Marshals Service (USMS) prisoners, federal inmates in the temporary custody of the USMS during transport and for criminal court appearances, in addition to detainees, and usually outnumbering them. ICE intended the 2019 NDS would provide the necessary guidance for approximately 45 facilities it had acquired by means of intergovernmental service agreements (IGSA) and had been operating already under the 2000 NDS, approximately 35 USMS facilities that ICE used and inspected against the 2000 NDS, plus approximately 60 facilities (both IGSA and USMS) which did not reach the threshold for ICE annual inspections—generally, those with an average daily population (ADP) of less than 10 detainees (ICE 2019a).

The NDS 2019 represented ICE's most significant departure from any of the preceding detention standards, and in my opinion, has created the most inconsistent conditions under which detainees are held today. In addition to the deference in treatment that INS had introduced in 2000, ICE eliminated 11 of 44 standards, a measure it minimized as merely a "consolidation", but in its place, ICE granted all those providers considerably more latitude in the operation of their facilities, particularly regarding healthcare, the clear consequence of which is even greater disparity in conditions of detention and far fewer protections for detainees in non-dedicated facilities than for those in dedicated facilities. ICE is not concerned, "These are facilities across the country where ICE's state and local law enforcement partners successfully manage their own [criminal] populations under federal, state, and local regulations (ICE 2019a)". Even if this were true, which I do not believe to be the case, applying correctional practices and then superimposing local healthcare policy on many of the facilities that are a part of a national system of immigration detention fails to protect civilly held people who are entitled to more as a matter of law.

As described by ICE, it appeared that the agency intended the 2011 (rev. 2016) PBNDS would replace the 2008 and 2011 Performance-Based National Detention Standards and that the 2019 NDS would replace the 2000 National Detention Standards; however, as

indicated in Table 2 that has not been the case. Instead, ICE renamed the 2000 NDS, the 2000 NDS for Non-Dedicated Facilities and continues to use it although other than the change in its name, it has not been revised since its 2000 release. Similarly, the 2008 PBNDS and 2011 PBNDS are still in use, as they were originally released. In mid-June 2021, there were 38 dedicated and 84 non-dedicated adult detention facilities in use. Not all dedicated facilities were assigned performance-based detention standards, 36 were and 2 were not. Not all non-dedicated facilities were assigned national detention standards, 63 were and 21 were not.

**Table 2.** Adult Detention Facilities by Type and Assigned Detention Standards (ICE 2021b).

| Type ICE Adult Facility | NDS 2000 | NDS 2019 | PBNDS 2008 | PBNDS 2011 | PBNDS 2011 (rev. 2016) | Totals |
|---|---|---|---|---|---|---|
| Dedicated (ICE only) Detention Facilities | | | | | | |
| CDF | | | | 3 | 11 | 14 |
| SPC | | | | | 5 | 5 |
| DIGSA | | 2 | | 2 | 15 | 19 |
| Subtotals | | 2 | | 5 | 31 | 38 |
| Non-dedicated (shared use) Detention Facilities | | | | | | |
| IGSA | 11 | 19 | 7 | 5 | 5 | 47 |
| USMS CDF | | 1 | 1 | | | 2 |
| USMS IGA | 27 | 5 | 3 | | | 35 |
| BOP | | | | | | 0 |
| Subtotals | 38 | 25 | 11 | 5 | 5 | 84 |
| Totals | 38 | 27 | 11 | 10 | 36 | 122 |

ICE occupied 122 facilities on 14 June 2021, of which 38 were dedicated and 84 were non-dedicated or shared use.

It is not unusual for an organization or professional association to promulgate more than one set of standards, when each set is tailored to a specific population to meet their unique needs. ICE understands this. ICE issued detention standards specifically for Family Residential Facilities in 2007 for that reason, and updated it in 2020, replacing one with other (ICE FRS 2020).

However, unlike the ACA and other professional organizations that replace older standards with newer ones when revised, ICE has kept the old and added each new set, assigning each facility one version or another. Now, ICE has five different sets, and all of them are still in use. More than confusing, it is unconscionable to detain the same population under appreciably different conditions, more so in order to qualify more facilities to house detainees, which is what ICE has done. Today, ICE assigns adult detainees to facilities operating under *five* different sets of expectations—2000 NDS, 2008 PBNDS, 2011 PBNDS, 2011 PBNDS (rev. 2016), *and* 2019 NDS—to maximize its bed capacity.

## 6. Government Oversight

Everyone is assured equal protection under the law. ICE's questionable use of detention standards, compounded by its inability to secure the facility operators' compliance with those standards, have been repeatedly scrutinized by the Government Accountability Office (GAO) and DHS Office of Inspector General (OIG), both of which are charged with oversight of federal agencies and focus on efficiency and integrity on behalf of the legislative and executive branches, respectively. To date, ICE appears to be neither deterred nor dissuaded.

In 2014, the GAO evaluated the three sets of detention standards that ICE had at that time and concluded employing more than one set of standards impeded ICE's ability to operate a uniformly effective and efficient system (GAO 2014). ICE disregarded its

advice. In 2016, the GAO determined similar practices impeded IHSC's efforts to collect information about on-site and off-site health care services and assess utilization (GAO 2016a). Again, ICE disregarded its advice. Also in 2016, the GAO addressed ICE's inability to utilize the correct version of each set of detention standards—the abbreviated version for under "72-h" facilities or the complete version for "over 72-h" facilities (GAO 2016b). ICE disregarded its advice.

In 2018, the DHS OIG concluded ICE's methods for monitoring facilities' compliance with their respective detention standards had failed and many of the deficiencies that it had identified were longstanding (OIG 2018). In 2019, the OIG probed further and found just 28 of the 106 contracts that it reviewed, approximately half of ICE's 206 contracts for beds at that time, included the Quality Assurance Surveillance Plan (QASP), a provision enabling ICE to impose financial penalties to ensure facilities met performance standards. The OIG determined where there was a QASP in place, ICE had imposed financial penalties on only two occasions despite numerous documented instances of facilities' failures to comply with detention standards. Instead, ICE issued waivers, exempting facilities with deficient conditions from complying with certain standards. The OIG discovered ICE also failed to issue written instruction to govern the waiver process, thereby enabling staff to continue to grant waivers without clear authority to do so (OIG 2019).

## 6.1. ICE's Oversight

The GAO and DHS OIG reports illustrate another reason that ICE should have just one set of detention standards—one set comprised solely of evidence-based practices—and it is this. ICE is unable to get most of the facilities it uses to comply with one set of standards; it is at least five times as unlikely that it will ever achieve compliance when there are five different sets of expectations in the field. There are several reasons why this is the case.

First, ICE's standards for acceptable and unacceptable performance (ICE 2021a) do not adequately address conditions that detainees encounter. Both acceptable and unacceptable are highly subjective terms. Not all deficiencies are equal. Frequency and severity vary—and no objective benchmarks are provided.

Additionally, ICE monitors detention facilities by several means. It assigns on-site agency monitors to its largest facilities. Independent reviews and fairly thorough inspections are also conducted at some facilities every several years by the Office of Detention Oversight, an independent office within ICE but outside of ERO. ICE also contracts with the Nakamoto Group Inc. to inspect most facilities annually. Those inspections were suspended through much of 2020 due to the pandemic then resumed remotely. Both oversight agencies and Congressional committees have been critical of ICE's contract management including its continued use of Nakamoto, and its inability to achieve better results over time. Chief among their concerns, no matter how poorly facilities perform, both the on-site agency monitors and the Nakamoto Group report that they Meet Standards. The GAO and OIG have also issued reports about ICE's failed oversight of detention operators, the most recent of which were published by the DHS OIG in June 2018 (OIG 2018), and the GAO in August 2020 (GAO 2020).

## 6.2. ICE Detention Today

Mid FFY2021, ICE had agreements to house adult detainees in 131 facilities in 38 states of which, 122 were in use. They are 38 dedicated detention centers housing ICE detainees only, and include five Service Processing Centers (SPCs),[6] 19 dedicated Intergovernmental Service Agreement (DIGSA) facilities,[7] and 14 Contract Detention Facilities (CDF),[8] all of which are supposed to comply with ICE Performance-based National Detention Standards (PBNDS). The remaining 84 facilities are non-dedicated, housing detainees, inmates and other prisoners, and include 47 Intergovernmental Service Agreement (IGSAs),[9] 35 U.S. Marshals Service Intergovernmental Agreements (USMS IGAs),[10] and two U.S. Marshals Service Contract Detention Facilities (USMS CDFs).[11] No detainees were assigned to a DOJ Bureau of Prison (BOP) facility at that time.[12] Most of these facilities are supposed

to comply with ICE's National Detention Standards. The USMS agreed to adopt the 2019 NDS at USMS CDF facilities but not at USMS IGA facilities; instead, ICE agreed to utilize the USMS contracts already in place with those providers. ICE also agreed to accept BOP standards when using its facilities.

Although dedicated detention facilities are supposed to comply with PBNDS detention standards and non-dedicated detention facilities, NDS detention standards, it does not always work out that way. Just these several detention practices—civilly held immigration detainees many of them comingled with criminally charged and convicted inmates, in over 100 correctional facilities, operating under five different sets of expectations, all of them based upon corrections policy and practice, in conditions more restrictive than many pre-trial prisoners are exposed—illustrate how insidious incarceration can be.

Currently, there is some discussion "on the Hill" about moving away from privately owned and operated correctional facilities altogether and using only those that are publicly owned and operated. Numerous studies by Congress and the White House have concluded most of the detention facilities that ICE uses are chronically deficient. Advocates point out, public or private, they are still correctional facilities, staffed with correctional personnel, operating pursuant to correctional detention standards, holding detainees in conditions as punitive as those in jails and prisons, perpetuating the belief that detainees are dangerous and should be punished. Some imagine readily available, non-secure settings appropriate for most civilly-held individuals, conveying the civil nature of their proceedings, the contributions they made already before their arrival, and their suitability to be our neighbors (Schriro 2009).

## 7. Detainee Healthcare: The Right to Receive Treatment

The case law is clear. Adequate healthcare is a fundamental right of the detained (Estelle v. Gamble 1976), and it cannot be conditioned upon the facility to which detainees are assigned (Cuoco v. Moritsugu 2000).[13] ICE must provide detainees with the actual care necessary to treat their medical conditions at every facility (Rosemarie M. v. Morton 2009). This can only occur when one clear set of expectations consistent with the corresponding case law is uniformly executed nationwide.

The overall responsibility for detainee healthcare rests with the Immigration Health Service Corps (IHSC) within Enforcement and Removal Operations (ERO). The IHSC serves as the medical authority for detainee healthcare issues, establishes the formulary, and oversees the financial authorization and payment for off-site specialty and emergency care services. The IHSC is also the healthcare provider at approximately half of 38 to 40 dedicated detention facilities and provides medical case management and oversight of the medical care administered by 84 non-IHSC providers at the other facilities (ICE IHSC 2021). Unlike correctional healthcare however, which is premised on the community standard of care, the IHSC deviates in its delivery, conditioning care on cost containment and anticipated time to removal or release, all too often delaying or denying care. Frequently occurring examples of IHSC's questionable decision-making include denials of corrective lenses and hearing aids to address vision and hearing impairments, dental cleanings within the first six months at a facility, and dental treatment for cavities—instead, detainees are redirected to the commissary to purchase "cheaters" regardless of their vision problem, and teeth requiring attention are extracted; cavities are not filled. Most physical ailments are treated with ibuprofen, and some mental health symptoms as well; there are no clinical services.

Although ICE's healthcare policy is established by IHSC, independently of ICE, IHSC is not responsible for healthcare outcomes. Instead, the delivery of detainee healthcare, and ultimately, detainees' health and safety, are the responsibility of each detention facility with which ICE contracts in accordance with that agreement which specifies in part, its assigned detention standards. This is an especially impactful provision at all the non-dedicated facilities where the state or local health department determines what is medically necessary.

### 7.1. Performance-Based National Detention Standards (PBNDS) for Dedicated Facilities

2008 PBNDS, 22 Medical Care, states, "All detainees shall have access to emergent, urgent, and non-emergent medical, dental, and mental health care within the scope of services provided by the Division of Immigration Health Services (ICE 2008c)".

*2011 PBNDS, 4.3 Medical Care*, states, "All detainees shall have access to appropriate and necessary medical, dental, and mental health care, including emergency services (ICE 2011c)".

2011 PBNDS (rev. 2016), 4.3 Medical Care, also provides, "All detainees shall have access to appropriate and necessary medical, dental, and mental health care, including emergency services (ICE 2016c)".

### 7.2. National Detention Standards for Non-Dedicated Facilities

*2000 NDS, Medical Care*, states, "All detainees shall have access to medical services that promote detainee health and general well-being (ICE 2002c)".

*2019 NDS, 4.3 Medical Care*, Policy, states, "All detainees shall have access to appropriate medical, dental, and mental health care, including emergency services (ICE 2019c)".

The inconsistencies in expectations and service delivery were especially apparent during the pandemic.

### 7.3. Pandemic Planning and Preparation

Pandemic planning and preparation are not new and there is no question as to its necessity. Over the course of DHS' 20-year history, the federal government has responded to the Severe Acute Respiratory Syndrome (SARS) between 2002 and 2004, H1N1 in 2009, Middle East Respiratory Syndrome (MERS) in 2012, Ebola between 2014 and 2016, Zika between 2015 and 2016, and the Coronavirus (COVID-19) since 2019 (Council on Foreign Relations 2021). Despite the continual threat each outbreak presents systemwide, ICE's Medical Care Detention Standards vary considerably in their responses to infectious disease and infection control.

### 7.4. Performance-Based National Detention Standards (PBNDS) for Dedicated Facilities

PBNDS 2008, PBNDS 2011, and PBNDS 2011 (rev. 2016) Detention Standard Medical Care considered Communicable Disease and Infection Control at length, and provide detailed instructions to identify and address tuberculosis, significant communicable diseases (the most commonly occurring, chicken pox, measles, mumps, whooping cough, and typhoid), and blood-borne pathogens (notably, hepatitis and HIV).

### 7.5. National Detention Standards for Non-Dedicated Facilities

NDS 2000 and 2019 NDS Detention Standards Medical Care are far narrower in their consideration of communicable diseases, contemplating just the identification of tuberculosis and only during the intake screening, although NDS 2019 is also the only Medical Care standard to reference CDC guidelines, including CDC Guidelines for Correctional Facilities, in its screening requirements for TB.

NDS 2000 also dedicated a section to HIV/AIDS, the only standard to address operational issues associated with HIV/AIDS when it was published. Although NDS 2000 is still in use, it has never been updated and this section is outdated and should be revised or removed.

Regarding the treatment all other Infectious and Communicable Diseases, both NDS 2000 and 2019 are quite terse. NDS 2000 states in its entirety, "[d]etainees diagnosed with a communicable disease shall be isolated according to local medical operating procedures". NDS 2019 directs, "[t]he facility will have written plans that address the management of infectious and communicable diseases, including testing, isolation, prevention, and education. This also includes reporting and collaboration with local or state health departments

in accordance with state and local laws and recommendations". It is up to the state or local health department to determine what this is.

ICE's decision that non-dedicated facilities are governed by state or local law is consequential when trying to assess the impact of COVID-19 on detainees in those facilities: in keeping with NDS 2000 and NDS 2019, each facility shall report positive test results—and deaths attributed to the coronavirus—according to that jurisdiction's policy or practice. As of June 2021, ICE reported detention facilities had administered 219,547 COVID-19 tests to detainees of which, 18,797 tests were positive for the coronavirus (8.5%) including nine patients known to have died and 851 patients currently in ICE custody, (ICE 2021b), a considerable number at a time that the percent of infected people in the community was at its lowest value in over a year (CDC 2021b). Studies show the actual number of COVID-19 detainee deaths nationwide may be as much as 5.5 times greater than reported by ICE due to jurisdictional differences in testing and reporting practices (Dolovich 2021).

### 8. ICE's Adaptation of CDC Interim Guidance on Management of Coronavirus Disease 2019 in Correctional and Detention Facilities

IHSC issued ICE's initial instructions to the field (ICE IHSC 2020). Thereafter, ERO released an Action Plan (ICE ERO 2020f) and then, Pandemic Response Requirements (PRR) (ICE ERO 2020a), to implement the CDC Interim Guidance (CDC 2020). ICE's earliest releases are especially revealing as to the dichotomy that differing detention standards created.

Interim Reference Sheet on 2019-Novel Coronavirus (COVID-19), Version 6.0. The first Interim Reference Sheet to be made available to the public is Version 6.0, on 6 March 2020, concerning CDC's expanded testing to include a wider group of symptomatic detainees ICE. IHSC's Sheet directed facility providers use their judgement to determine whether patients should be tested. It also "strongly encouraged" them to test for other causes of respiratory illness such as influenza (ICE IHSC 2020).

Coronavirus Disease 2019 (COVID-19) Action Plan, Revision 1. The next publicly available document and ERO's first release is its COVID-19 Action Plan, Revision 1, dated 27 March 2020 (ICE ERO 2020f). It was ICE's most comprehensive effort to date to mitigate risk of infection and transmission among detainees and staff but applied *only* to IHSC-staffed and non-IHSC-staffed, ICE-dedicated facilities.

Intergovernmental partners and non-dedicated facilities were instructed to take their directions from their local, state, tribal, territorial, and federal public health authorities, although it recommended that they consider the dedicated facilities' instructions as "best practices". It was one of the earliest and clearest demarcations in ICE's expectations for the field's response to COVID-19: Dedicated facilities must comply, non-dedicated facilities may. In fact, few did.

COVID-19 Pandemic Response Requirements. ERO released COVID-19 Pandemic Response Requirements (PRR), Version 1, on 10 April 2010 (ICE ERO 2020a), the first of six, addressing an agency-wide healthcare crisis with some requirements for dedicated detention facilities, and others for non-dedicated facilities, and a statement of sorts for all facilities.

In June 2020, PRR Version 2 attempted to address the considerable confusion—and criticism—that its facility-specific approach to a nationwide threat had generated, now insisting ERO's PRR establishes mandatory requirements, as well as best practices, for *all* its detention facilities in response to COVID-19 (ICE ERO 2020b). The DHS Office of Inspector General (OIG) disagreed with ICE's assertion that it had issued universal expectations for all facilities in its in June 2020 report about detention facilities' early experiences with COVID-19), reiterating ICE had provided guidance regarding COVID-19, but only dedicated detention facilities must comply (OIG 2020).

PRR Version 2 brought to light another disparity in ICE's detention management. Not all its agreements with facility operators contained compliance measures, and where there were provisions, they varied by contract in their consequences, and others had no provisions for penalties. Specific to the pandemic, differences in the facilities' provisions

to impose sanctions for non-compliance with the PRR varied considerably and none of the dedicated facilities without a certain mechanism, a quality-assurance surveillance plan (QASP), could be penalized (OIG 2020). In another report by the DHS OIG just the year before, and referenced above, it found only a few of the contracts it had reviewed included a QASP, and ICE had exercised this provision on only two occasions (OIG 2019).

PRR Version 3 issued in July 2020 (ICE ERO 2020c), PRR Version 4 issued in September 2020 (ICE ERO 2020d), PRR Version 5 issued in October 2020 (ICE ERO 2020e), and PRR Version 6 issued in March 2021 (ICE ERO 2021) continued to differentiate detention operators' responsibilities by facility type.

**9. ICE's Pandemic Plan: Feedback from Federal Oversight Agencies**

Both the GAO and the DHS OIG have released reports about ICE's readiness for and response to the pandemic.

Department of Homeland Security (DHS) Office of Inspector General. In April 2020, the DHS OIG surveyed 196 detention facilities in use at that time about their experiences and challenges managing COVID-19; 188 facility operators responded representing 31 dedicated and 157 non-dedicated facilities, of which only 18 dedicated facilities were IHSC-staffed. Overall, 93% (175) of the facilities reported they were prepared to handle COVID-19 (OIG 2020). Generally, respondents stated they had adequate supplies for detainees to mitigate the spread of COVID-19. Specifically, 89% (168) said they had enough masks for detainees who exhibited COVID-19 symptoms or tested positive for COVID-19. About 90% (170) of facilities reported having enough liquid soap for detainees but more than one-third (69) reported not having enough hand sanitizer for their use.

There were demonstrable differences however, in readiness by facility type. For example, 85% of dedicated facilities (26 of 31) had on-site testing capacity compared to 54% of non-dedicated facilities (84 of 157). The disparity and its impact are significant: The ability to test on-site frequently determines whether detainees are tested at all—77% (24 of 31) of dedicated facilities reported testing detainees for potential COVID-19, whereas only 20% (32 of 157) of non-dedicated facilities reported doing so (ICE 2020b).

Quite a few facilities also reported significant limitations due to their physical space, its configuration and size. Of note, 11% (21) did not have the capacity to quarantine or isolate detainees who exhibited suspected COVID-19 symptoms, 12% (23) could not quarantine or isolate a detainee who had tested positive for COVID-19; and 29% (55) did not have negative pressure ventilation rooms to isolate airborne infections. Another one-third (62) had only one or two negative pressure rooms in their facilities.

Again, survey results conveyed the disparity between dedicated and non-dedicated facilities. Every dedicated facility (31) reported being able to quarantine or isolate detainees with confirmed cases of COVID-19, whereas 15% (21 of 157) of non-dedicated facilities reported they could not, and all but one dedicated facility had negative pressure ventilation rooms while 34% (54 of 165) of non-dedicated facilities did not.

As mentioned in the previous section, the OIG also took note that ICE had provided guidance regarding COVID-19 to all the facilities, much of which was applicable only to dedicated facilities and facilities with IHSC staff, and non-dedicated facilities and those without IHSC staff—the majority—were not obligated to comply. The artificial line that ICE created as to facilities' accountability also affected its efforts to communicate effectively with the field. The OIG determined about 83% (156) of facilities had received COVID-19 guidance from ICE headquarters and 75% (141) had received guidance from IHSC. Responses regarding the receipt of guidance differed however, between dedicated and non-dedicated facilities. For example, every dedicated facility reported it had received guidance from ICE regarding COVID-19, whereas almost 20% (32) of non-dedicated facilities reported they had not. Similarly, all but one dedicated facility reported receiving IHSC guidance, while 27% (43) of non-dedicated facilities reported they did not. It is difficult for the non-dedicated facilities to consider information from ERO as best practices when they were not received.

In September 2021, the DHS OIG released its assessments of nine detention facilities' responses to COVID-19 (OIG 2021). The OIG conducted unannounced remote inspections in response to congressional requests for a more in-depth review than the year before to determine whether ICE effectively controlled COVID-19 and adequately safeguarded the health and safety of detainee and detention staff. The areas that the OIG considered included maintaining adequate supplies of personal protective equipment, enhanced cleaning, and proper screening for new detainees and staff. The OIG identified a number of areas where the facilities struggled to properly manage the health and safety of detainees. For example, they observed instances where staff and detainees did not consistently wear face masks or socially distance. They also noted that some facilities did not consistently manage sick calls and did not regularly communicate with detainees about their COVID-19 test results. Although the OIG found that ICE was able to decrease the detainee population to help mitigate the spread of COVID-19, information about their transfers was limited. Its staff also found that testing of both detainees and staff was insufficient, and that ICE headquarters generally did not provide effective oversight of the facilities during the pandemic. Overall, the OIG concluded, ICE must resolve these issues to ensure it can meet the challenges of the COVID-19 pandemic, as well as future pandemics.

The Government Accounting Office. In June 2021, the GAO released a report summarizing its examination of ICE's policies and procedures for responding to COVID-19 in the field and how they were implemented at six facilities; ICE's mechanisms for conducting oversight of COVID-19-related health and safety measures; and ICE's data on COVID-19 cases and identified high-risk health factors among detainees, between January 2020 and March 2021 (GAO 2021). The study had been requested by unspecified Congressional committees. GAO staff reviewed ICE documents and interviewed ICE officials at headquarters and select facility operators between May 2020 and June 2021 about initial ERO communication, interim guidance and policy documents, detainee intake screening and testing, the identification of high-risk detainees, quarantine and isolation, hygiene and PPE supplies, cleaning and disinfection, social distancing and education efforts, and visitation procedures. The report summarized the interviews and surveys upon which ERO relied to monitor facilities' COVID-19 activities remotely during the pandemic. The GAO staff did not consider detainee grievances, formulate opinions, or make any recommendations, as it often does.

## 10. Harm and Risk Mitigation

The purpose of a viable custody classification system is to ensure safety and security and contribute to orderly facility operations, by separating and managing detainees based on verifiable and documented data. A thorough screening by qualified personnel during the admissions process is also crucial for the identification of individuals for whom detention would be detrimental to their health and/or wellbeing therefore merit modification of the conditions of confinement, transfer to a suitable facility. Rarely are individuals considered for release, although policy does not prevent it.

In CY2020, ICE tested 80,200 of 137,749 detainees (58%) for COVID-19 and recorded 8622 positive cases (11%) at over 100 immigration detention facilities. Approximately, 30% (2566) of positive COVID-19 cases occurred at the 18 IHSC-staffed facilities whereas the remaining 70% (6056) of cases occurred at facilities operated by contract medical staff or the local health authority. Of the detainees who tested positive for COVID-19 in 2020, approximately 89% (7687) were exposed to the virus while in ICE custody, whereas 5% (435) were exposed before contact with ICE. Eight detainees died while in ICE custody as a result of COVID-19 (GAO-21-414). At least several more are believed to have died in 2020 shortly after their release or removal (Smart et al. 2021). The data strongly suggest adequate screening and testing for COVID-19, correct and consistent use of PPE, sufficient space for quarantine, and information and access to vaccination could have slowed the spread of COVID-19 and saved lives.

## 11. ICE Risk Assessment

The mission of the DOJ National Institute of Corrections (NIC) is to advance public safety by shaping and enhancing correctional policies and practices. The NIC created an objective classification system for the INS at its request to ensure every detainee is placed in the appropriate category of risk—low, medium–low, medium–high, or high—and physically separated from detainees in other custody levels in the least restrictive housing consistent with facility safety and security. To do so reduces lower custody detainees' exposure to any potential physical and psychological danger that higher custody detainees may pose (ICE 2002a). The risk assessment process in ICE's five Custody Classification detention standards is also used to ascertain detainees' suitability for release and the conditions, if any, that may be warranted to ensure their compliance with court appearances.

NDS 2000 Detention Standard Detainee Classification System (ICE 2002b) is the most comprehensive of the five standards. As written, custody staff *in consultation with* medical and mental health clinicians can consistently produce highly accurate assessments of risk and need for special housing.

PBNDS 2008 (ICE 2008b), PBNDS 2011 (ICE 2011b), and PBNDS 2011 (rev. 2016) (ICE 2016b) Detention Standard 2.2 Custody Classification lack some of its specificity however there is sufficient instruction that staff can accurately assess detainee risk. The primary difference between PBNDS 2008, 2011, and 2011 (rev. 2016) and NDS 2000 is that these three rely primarily on detention officers to assess the detainees thus there must be adequate training and continual oversight by healthcare personnel to achieve a good result.

Additionally, the NDS 2000, PBNDS 2008, PBNDS 2011, and PBNDS 2011 standards (rev. 2016) include a user's manual and assessment forms or worksheets to promote consistently reliable outcomes. NDS 2000 also includes a monitoring instrument, the Primary Assessment Form, to assess each facility's compliance with the policy.

NDS 2019 Standard Detainee Classification System (ICE 2019b) is by far the least likely to achieve a good result. It has few instructions and no worksheets or forms. Detention staff is expected to complete assessments without assistance or support. NDS 2000 with attachments is 33 pages whereas, NDS 2019 is just three pages. Sometimes size matters—this is one of those times. Since the advent of COVID-19, reliable risk assessments are more consequential than ever.

### 11.1. ICE Special Vulnerabilities and Management Concerns

The NIC recognized that some detainees have special vulnerabilities and/or management concerns and there also should be provision in the classification process for their identification to inform housing assignments and accommodate certain handicapping conditions.

NDS 2000 Detainee Classification System and PBNDS 2008 Classification System identified only several Special Management Concerns—psychological impairments, mental deficiency, substance abuse, and detainees with medical problems or physical impairments. PBNDS 2011 2.2 Custody Classification System, PBNDS 2011 (rev. 2016) Custody Classification System, and NDS 2019 Custody Classification recognized quite a few special vulnerabilities—the elderly, those who are pregnant or nursing; those with serious physical or mental illness, or other disabilities; those who would be susceptible to harm in general population related to their sexual orientation or gender identity; and victims of sexual assault, torture, trafficking, or abuse. Having reviewed hundreds of custody classification worksheets in numerous facilities over the past ten years however, I can attest most Intake Officers do not complete this section and I do not believe they have had the training to do so correctly if they were directed. That so few are completed, or completed correctly, also underscores the need for training and continual supervision.

### 11.2. Prosecutorial Discretion

As a matter of policy, ICE has always had prosecutorial discretion to release individuals with serious medical conditions and individuals who are vulnerable to medical

harm. The release of individuals with special vulnerabilities from immigration detention is authorized under a range of statutory and regulatory provisions, notably INA §§ 212(d)(5), 235(b), 236, and 241, and 8 C.F.R. §§ 1.1(q), 212.5, 235.5, and 236.(b), and even individuals held under mandatory detention pursuant to INA §§ 236(c). As a matter of practice, ICE is usually unwilling to do so, even when ordered by the Court.

In 2000 when NDS 2000 was issued, there was no ATD program to which detainees with special vulnerabilities and certain management concerns could be referred. This is no longer the case. ICE has operated ATD programs since 2004 (ICE 2021d). In July 2021, ICE updated its policy on the arrest and detention of pregnant, postpartum, and nursing women (ICE 2021c). The new policy directs ICE and CBP to limit the arrest of pregnant and nursing women, and it establishes new guidelines on how to treat them if they are detained. It addresses only two of many at-risk categories of detainees identified by the CDC and the Court who are at risk of serious illness or death from the coronavirus unless released.

## 12. CDC Guidance, Underlying Medical Conditions Associated with High Risk for Severe COVID-19

The CDC identified a number of categories of people more likely to get severely ill from COVID-19 (CDC 2021a). With regard to adults, the CDC considered both at-risk adults of any age and older adults.

Adults of Any Age. Adults of any age with the following conditions can be more likely to get severely ill from COVID-19: cancer, chronic kidney disease, chronic lung diseases including COPD (chronic obstructive pulmonary disease), asthma (moderate-to-severe), interstitial lung disease, cystic fibrosis, and pulmonary hypertension, dementia or other neurological conditions, diabetes (type 1 or type 2), Down syndrome, heart conditions such as heart failure, coronary artery disease, cardiomyopathies or hypertension, HIV infection, immunocompromised state, liver disease, overweight and obesity, pregnancy, sickle cell disease or thalassemia, current and former smokers, recipients of a solid organ or blood stem cell transplant, stroke or cerebrovascular disease, and substance use disorders.

Older Adults. Older adults are more likely to get severely ill from COVID-19. More than 80% of COVID-19 deaths occur in people over age 65, and more than 95% of COVID-19 deaths occur in people older than 45. Additionally, people exposed to long-standing system health and social inequities including many racial and ethnic minority groups and people with disabilities are more likely to both get COVID-19 and have worse outcomes.

## 13. Fraihat Risk Factors

In March 2020, attorneys on behalf of Plaintiffs Fraihat et al., sought relief for detained people with certain risk factors including those who are older, pregnant, or who have underlying medical conditions and are at a heightened risk of serious illness, life-altering complications, and death from COVID-19 (Fraihat v. ICE 2020). Plaintiffs argued successfully that ICE's responses to COVID-19 and its inadequate healthcare system will not protect people with risk factors. In April 2020, the Court ordered ICE review for release every person in the class.

Since then, all immigration detention centers are required to evaluate every new admission within five days of admission, to identify the presence of factors that may place a detainee at higher risk for severe illness due to COVID-19-related risk factors or disabilities. Based on the *Fraihat* ruling and related CDC guidance, ERO's PRR now requires every facility identify all the detainees with these chronic health conditions—cancer, chronic kidney disease, chronic obstructive requires pulmonary disease, Down syndrome, weakened immune system, overweight and obesity, serious heart conditions, including heart failure, coronary artery disease and cardiomyopathies, sickle cell disease, type one and type two diabetes mellitus, asthma, cerebrovascular disease, cystic fibrosis, hypertension or high blood pressure, neurologic conditions, including dementia, liver disease, pulmonary fibrosis, smoking, and thalassemia (ICE ERO 2020b, 2020c, 2020d, 2020e, 2021).

According to ICE data, in CY2020 facility medical staff determined 14,728 detainees had one or more conditions that placed them at high risk for severe illness due to COVID-19 of which, ICE released 5801 detainees (39%) from custody, removed another 5432 high-risk detainees from the United States (37%), and continued to detain 3487 (24%) as of the end of CY2020 (GAO 2021).

ICE needs to do more, now. As of 30 March 2021, 528 high-risk detainees have tested positive for COVID-19 (GAO 2021). *Fraihat* demonstrates just how inadequate ICE's classification policy and practice are. Few, if any, of the detainees with conditions that placed them at high risk for severe illness due to COVID-19 were known to ICE, and it is unlikely that ICE would have exercised the discretion it already had to release any of them.

## 14. Is the Past Prologue? Summary, Conclusions, and Recommendations

ICE does not handle infectious and communicable diseases well. Every year, detention facilities encounter detainees with measles, mumps, and chicken pox, and the prospect of significant consequences for some. Nevertheless, with each outbreak, impacted facilities do the same things the same ways. They lock down entire housing units, and occasionally the entire building, even when a simple screening for prior infection and/or verification of inoculation is all that it needed to return many of those detainees to general population. Some lockdowns have been so large and lasted so long that court runs have been cancelled, attorney visits forfeited, and access to outdoor recreation, the legal orientation program, and the law library were suspended. These outbreaks, always addressed the same way, occur so often that what were once questionable practices are widely accepted now as best practices for handling infectious and communicable diseases. It should not come as a surprise then, when IHSC and ERO directed the field offices and detention facilities in early 2020 to review their communicable disease and infection control plans in anticipation of the pandemic, it only served to fortify bad practices already entrenched systemwide. They thought they knew everything that they would need to know: 'it will pass'.

Although all detainees are always in the custody of ICE, their access to personal protective equipment, sanitation and hygiene supplies, testing and vaccine, adequate conditions for quarantine, as well as routine and emergency healthcare is dependent upon the facilities to which ICE has assigned them; and, as was demonstrated with respect to access to test kits and kit processing—the ability to test increases the ability to identity and address infected detainees—facilities' access to essential supplies, staff and space also varies considerably, and to the detriment of the individuals for whom ICE is responsible.

ICE's failures to take measures to mitigate the harm to which detainees continue to be exposed, and the heightened danger to which ICE has exposed at-risk detainees throughout the pandemic, must be addressed.

These are several of the ways that measurable improvements can be realized.

One, ICE is a federal enforcement agency. ICE should ensure all its policies and practices comply with immigration case law, are uniform and uniformly enforced, and every person in its custody receives equal treatment.

Two, to that end, ICE should operate a unified system, a system with one set of standards expressing the highest expectations and a continuum of control ranging from no supervision to detention, the premise being most require little or no supervision.

Three, ICE should decriminalize its policies and procedures, its facilities and ATD programs. ICE should discontinue use of jails and prisons, especially non-dedicated facilities where detainees are collocated and comingled with correctional populations, as well as correctional supervision strategies and correctional policies and procedures to the greatest extent practicable, as quickly as practicable.

Four, decisions as to detainees' placement along the continuum of control should be based on objective assessments of risk and a thorough identification of vulnerabilities. ICE should retain expert assistance and revise its classification process and also, add instruments to identify vulnerable persons and accurately identify security risk groups and members.

Five, upon the revision of risks and needs assessments, all detainees should undergo reclassification and low risk and at-risk detainees reconsidered for release under the least restrictive means. Also, there always should be sufficient personnel qualified to assess risk and identify vulnerabilities and make timely referrals at all facilities.

Six, mandatory detention should reflect real risk, and "in the custody of" should be expanded to include alternatives-to-detention programs. ICE should conduct a review of detainees' custody classification files currently held under mandatory detention provisions to identify anyone who may be detained erroneously under its provisions and arrange for their release. Depending upon the outcome of the review, additional training of Intake staff, and revision of custody classification detention standard and/or the INA § 236(c) and § 235(b) may be required as well.

Seven, detainee healthcare should meet or exceed the community standard of care at every location. Detainees' access to healthcare should not be conditioned on county or state policy. Every facility that ICE uses to detain individual in its custody must also be capable of complying fully with CDC Guidance. It is essential that IHSC establishes a universal standard for detainee healthcare for all facilities.

Eight, IHSC should conduct an immediate review of every facility to determine what levels and kinds of healthcare and which handicapping conditions cannot be accommodated. ICE must ensure anyone who cannot be accommodated at their current location is relocated or released immediately.

Nine, ICE is better positioned to act in the event of a pandemic than any of the detention facilities with which it contracts or the communities in which those facilities are located. For planning purposes, ICE should assume responsibility for the nation's immigration detention system's pandemic preparedness and response. It must ensure all detainees have timely access to requisite supplies and equipment, space for medical treatment and isolation, medicine, medical personnel, all necessary components of routine and emergency medical services, all of the time and at every location.

Ten, every facility that ICE uses for detention should be capable of complying fully with ICE's current detention standards, and upon revision thereof, one set of detention standards that complies with the case law. When that occurs, ICE should discontinue the use of any facility or facility provider that cannot meet these requirements. Until then, ICE should not issue any variances but for temporary conditions that can be readily and timely resolved.

**Funding:** This research received no external funding.

**Institutional Review Board Statement:** Not applicable.

**Informed Consent Statement:** Not applicable.

**Data Availability Statement:** Data available in a publicly accessible repository that does not issue DOIs. Publicly available datasets were analyzed in this study. This data can be found at: https://www.ice.gov/doclib/detention/FY21-detentionstats.xlx, accessed on 21 September 2021.

**Conflicts of Interest:** The author declares no conflict of interest.

## Notes

[1]    Unless noted otherwise, all data is presented by U.S. Federal Fiscal Year (FFY), 1 October–30 September.

[2]    ICE identified 28% more detainees subject to mandatory detention (70%) that it designated risk levels 1, 2, and 3 (42%).

[3]    ICE determines the threat level by the criminality of a detainee, including the recency of the criminal behavior and its severity. A detainee may be graded on a scale of one to three with one being the highest severity. When a detainee has no criminal convictions, s/he shall be classified as "No ICE Threat Level".

[4]    The average daily population (ADP) FFYTD on 14 June 2021, by ICE Threat Level.

[5]    Typically, this population consists of pre-trial inmates and inmates sentenced in a state court to a year or less.

[6]    The SPC (Service Processing Center) is a facility owned by the government and staffed by a combination of federal and contract employees.

7    The DIGSA (Dedicated Intergovernmental Service Agreement) is a publicly owned facility operated by state/local government(s), or private contractors, in which ICE contracts to use all bed space via a Dedicated Intergovernmental Service Agreement; or facilities used by ICE pursuant to Inter-governmental Service Agreements. Typically, the latter are operated by private contractors pursuant to their agreements with local governments.

8    The CDF (Contract Detention Facility) is owned and operated by a private company and contracts directly with ICE. A CDF houses only ICE detainees. Note: ICE no longer identifies the CDF on its website as one of the types of facilities it uses but lists 14 CDFs in its report.

9    The IGSA (Intergovernmental Service Agreement) is a publicly owned facility operated by state/local government(s), or private contractors, in which ICE contracts for bed space via an Intergovernmental Service Agreement; or local jails used by ICE pursuant to Inter-governmental Service Agreements, which house both ICE and non-ICE detainees, typically county prisoners awaiting trial or serving short sentences, but sometimes also USMS prisoners.

10    The USMS IGA (USMS Intergovernmental Agreement) is a facility where ICE agrees to utilize an already established US Marshals Service contract.

11    The USMS (United States Marshals Service) is a facility primarily contracted with the USMS for housing of USMS detainees, in which ICE contracts with the USMS for bed space.

12    The BOP (Federal Bureau of Prisons): a facility operated by the Federal Bureau of Prisons for federal inmates.

13    ICE must not disregard excessive risk to a detainee's health or safety at any facility.

## References

### Cases

Bell v. Wolfish, 441 U.S. 520, 535 (1979).

Biden, President of U.S.; et al. v. Texas; et al., 594 U.S. 21A21 (2021).

City of Revere v. Massachusetts General Hospital, 463 U.S. 239 (1983).

Cuoco v. Moritsugu, 222 F.3d. 99, 107 (2nd Cir., 2000).

Edwards v. Johnson, 209 F.3d 772, 778 (5th Cir. 2000).

Estelle v. Gamble, 429 U.S. 97, 104 (1976).

Fong Yue Ting v. United States, 149 U.S. 698, 728-30 (1893).

Fraihat et al. v. ICE et al., Case No. 19-cv-01546-JGB(SHKx), Mar. 24, 2020.

Helling v. McKinney, 509 U.S. 25, 31-32 (1993).

Jones v. Blanas, 393F. 3d. 918, 933-34, (9th Cir. 2004), cert denied, 546 U.S. 820 (2005).

Rosemarie M. v. Morton, 671 F. Supp. 2d 1311, 1313 (M.D. Fla. 2009).

United States v. Solano, 41 U.S. 739, 747 (1987).

Wilson v. Seiter, 501 U.S. 294, 303 (1991).

Wong Wing v. United States, 163 U.S. 228, 237-38 (1896).

Youngberg v. Romeo, 457 U.S. 307, 321-32, (1982).

Zadvydas v. Davis, 533 U.S. 678, 609 (2001).

### Statutes

Immigration and Naturalization Act 8 CFR § 236(c) and § 235(b).

Public Health Service Act, 42 U.S.C. §§ 265 and 268.

American Correctional Association. 2004. *Performance-Based Standards for Adult Local Detention Facilities*, 4th ed. Alexandria: American Correctional Association. Available online: https://www.aca.org/ACA_Prod_IMIS/ACA_Member/Standards_and_Accreditation/StandardsInfo_Home.aspx?WebsiteKey=139f6b09-e150-4c56-9c66-284b92f21e51&hkey=7c1b31e5-95cf-4bde-b400-8b5bb32a2bad&New_ContentCollectionOrganizerCommon=2#New_ContentCollectionOrganizerCommon=2 (accessed on 1 July 2021).

American Correctional Association. 2016. Standards Supplement. Available online: https://www.aca.org/ACA_Prod_IMIS/ACA_Member/Standards_and_Accreditation/StandardsInfo_Home.aspx?WebsiteKey=139f6b09-e150-4c56-9c66-284b92f21e51&hkey=7c1b31e5-95cf-4bde-b400-8b5bb32a2bad&New_ContentCollectionOrganizerCommon=2#New_ContentCollectionOrganizerCommon=2 (accessed on 1 July 2021).

Anonymous. 2021. Chapter 4: Conditions of Confinement, COVID-19, and the CDC. *Harvard Law Review* 134: 2233–56. Available online: https://harvardlawreview.org/2021/04/conditions-of-confinement-covid-19-and-the-cdc/ (accessed on 12 April 2021).

Bowling, Ben, and Sophie Westenra. 2018. Theoretical Criminology, 'A really hostile environment', Adiaphorization, global policing and the crimmigration control system. *Sage Journal* 24: 163–83. [CrossRef]

Centers for Disease Control and Prevention. 2019. Mission. Available online: https://www.cdc.gov/about/organization/mission.htm (accessed on 13 May 2019).

Centers for Disease Control and Prevention. 2020. Interim Guidance on Management of Coronavirus Disease 2019 (COVID-19) in Correctional and Detention Facilities. Available online: https://www.cdc.gov/coronavirus/2019-ncov/community/correction-detention/guidance-correctional-detention.html (accessed on 23 March 2020).

Centers for Disease Control and Prevention. 2021a. COVID-19 Medical Conditions, Updated May 13. Available online: https://www.cdc.gov/coronavirus/2019-ncov/need-extra-precautions/people-with-medical-conditions.html (accessed on 13 May 2021).

Centers for Disease Control and Prevention. 2021b. COVID Data Tracker Weekly Review. July 23. Available online: https://www.cdc.gov/coronavirus/2019-ncov/covid-data/covidview/index.html (accessed on 23 July 2021).

Cole, David. 2014. *The Difference Prevention Makes: Regulating Preventive Justice*. Washington, DC: Georgetown University Law Center, Crim. L. & Phil. [CrossRef]

Council on Foreign Relations. 2021. Major Epidemics of the Modern Era, 1899–2021. Available online: https://www.cfr.org/timeline/major-epidemics-modern-era (accessed on 2 July 2021).

Dolovich, Sharon. 2021. COVID Behind Bars Data Project. UCLA COVID Behind Bars Data Project. Available online: https://uclacovidbehindbars.org (accessed on 8 July 2021).

Goffman, Erving. 1961. *On the Characteristics of Total Institutions in Asylums: Essays on the Social Situation of Mental Patients and other Inmates*. Garden City: Anchor Books.

Government Accountability Office. 2014. Immigration Detention, Additional Actions Needed to Strengthen Management and Oversight of Facility Costs and Standards. GAO-15-153. Available online: https://www.gao.gov/products/GAO-15-153 (accessed on 10 October 2014).

Government Accountability Office. 2016a. Immigration Detention, Additional Actions Needed to Strengthen Management and Oversight of Detainee Medical Care. GAO-16-231. Available online: https://www.gao.gov/assets/680/675484.pdf (accessed on 29 February 2016).

Government Accountability Office. 2016b. Immigration Detention, Additional Actions Needed to Strengthen DHS Management of Short-Term Holding Facilities. GAO-16-514. Available online: https://www.gao.gov/assets/680/677484.pdf (accessed on 26 May 2016).

Government Accountability Office. 2020. Immigration Detention: ICE Should Enhance Its Use of Facility Oversight Data and Management of Detainee Complaints. GAO-20-596. Available online: https://www.gao.gov/products/gao-20-596 (accessed on 19 August 2020).

Government Accountability Office. 2021. Immigration Detention, ICE Efforts to Address COVID-19 in Detention Facilities. GAO-21-414. Available online: https://www.gao.gov/assets/gao-21-414.pdf (accessed on 30 June 2021).

Immigration and Customs Enforcement. 2002a. 2000 National Detention Standards for Non-Dedicated Facilities. Available online: https://www.ice.gov/detain/detention-management/2000 (accessed on 11 February 2002).

Immigration and Customs Enforcement. 2002b. 2000 INS Detention Standard, Detainee Classification System. Available online: https://www.ice.gov/doclib/dro/detention-standards/pdf/classif.pdf (accessed on 11 February 2002).

Immigration and Customs Enforcement. 2002c. 2000 INS Detention Standard, Medical Care. Available online: https://www.ice.gov/doclib/dro/detention-standards/pdf/medical.pdf (accessed on 11 February 2002).

Immigration and Customs Enforcement. 2008a. 2008 Operations Manual ICE Performance-Based National Detention. Available online: https://www.ice.gov/detain/detention-management/2008 (accessed on 11 March 2021).

Immigration and Customs Enforcement. 2008b. 2008 Performance-Based National Detention Standard, ICE/DRO Detention Standard, Classification System. Available online: https://www.ice.gov/doclib/dro/detention-standards/pdf/classification_system.pdf (accessed on 11 March 2021).

Immigration and Customs Enforcement. 2008c. 2008 Performance-Based National Detention Standard, ICE/DRO Detention Standard, Medical Care. Available online: https://www.ice.gov/doclib/dro/detention-standards/pdf/medical_care.pdf (accessed on 11 March 2021).

Immigration and Customs Enforcement. 2011a. 2011 Operations Manual ICE Performance-Based National Detention Standards. Available online: https://www.ice.gov/detain/detention-management/2011 (accessed on 11 March 2021).

Immigration and Customs Enforcement. 2011b. 2011 Performance-Based National Detention Standard, 2.2 Custody Classification System. Available online: https://www.ice.gov/doclib/detention-standards/2011/2-2.pdf (accessed on 11 March 2021).

Immigration and Customs Enforcement. 2011c. 2011 Performance-Based National Detention Standard, 4.3 Medical Care. Available online: https://www.ice.gov/doclib/detention-standards/2011/4-3.pdf (accessed on 11 March 2021).

Immigration and Customs Enforcement. 2016a. 2011 Operations Manual ICE Performance-Based National Detention Standards (rev. 2016). Available online: https://www.ice.gov/detain/detention-management/2011 (accessed on 11 March 2021).

Immigration and Customs Enforcement. 2016b. 2011 Performance-Based National Detention Standard (rev. 2016), 2.2 Custody Classification System. Available online: https://www.ice.gov/doclib/detention-standards/2011/2-2.pdf (accessed on 11 March 2021).

Immigration and Customs Enforcement. 2016c. 2011 Performance-Based National Detention Standard (rev. 2016), 4.3 Medical Care. Available online: https://www.ice.gov/doclib/detention-standards/2011/4-3.pdf (accessed on 11 March 2021).

Immigration and Customs Enforcement. 2019a. 2019 National Detention Standards for Non-Dedicated Facilities. Available online: https://www.ice.gov/detain/detention-management/2019 (accessed on 11 March 2021).

Immigration and Customs Enforcement. 2019b. 2019 National Detention Standard, 2.2 Custody Classification System. Available online: https://www.ice.gov/doclib/detention-standards/2019/2_2.pdf (accessed on 11 March 2021).

Immigration and Customs Enforcement. 2019c. 2019 National Detention Standard, 4.3 Medical Care. Available online: https://www.ice.gov/doclib/detention-standards/2019/4_3.pdf (accessed on 11 March 2021).

Immigration and Customs Enforcement. 2019d. ICE Detention Statistics, Facilities Data, EOFY2019. Available online: https://www.ice.gov/doclib/detention/FY19-detentionstats.xlsx (accessed on 7 October 2019).

Immigration and Customs Enforcement. 2020a. ICE Detention Statistics, Facilities Data, EOFY2020. Available online: https://www.ice.gov/doclib/detention/FY20-detentionstats.xlsx (accessed on 30 September 2020).

Immigration and Customs Enforcement. 2020b. ICE ERO FY2020 Achievements. Available online: https://www.ice.gov/features/ERO-2020/feature (accessed on 1 March 2021).

Immigration and Customs Enforcement. 2021a. ICE Detention Statistics Facilities Data, FY21 YTD. Available online: https://www.ice.gov/doclib/detention/FY21-detentionstats.xlx (accessed on 24 June 2021).

Immigration and Customs Enforcement. 2021b. COVID-19 ICE Detainee Statistics by Facility as of June 30, 2021. Available online: https://www.ice.gov/coronavirus#detStat (accessed on 30 June 2021).

Immigration and Customs Enforcement. 2021c. ICE Directive 11032.4, Identification & Monitoring of Pregnant, Postpartum or Nursing Individuals. Available online: https://www.ice.gov/doclib/detention/11032.4_IdentificationMonitoringPregnantPostpartumNursingIndividuals.pdf (accessed on 1 July 2021).

Immigration and Customs Enforcement. 2021d. History of ICE. Available online: https://www.ice.gov/features/history (accessed on 26 January 2021).

Immigration and Customs Enforcement, Enforcement and Removal Operations. 2020a. COVID-19 Pandemic Response Requirements, Version 1. Available online: https://www.ice.gov/doclib/coronavirus/eroCOVID19responseReqsCleanFacilities-v1.pdf (accessed on 10 April 2020).

Immigration and Customs Enforcement, Enforcement and Removal Operations. 2020b. COVID-19 Pandemic Response Requirements, Version 2. Available online: https://www.ice.gov/doclib/coronavirus/eroCOVID19responseReqsCleanFacilities-v3.pdf (accessed on 22 June 2020).

Immigration and Customs Enforcement, Enforcement and Removal Operations. 2020c. COVID-19 Pandemic Response Requirements, Version 3. Available online: https://www.ice.gov/doclib/coronavirus/eroCOVID19responseReqsCleanFacilities-v4.pdf (accessed on 28 July 2020).

Immigration and Customs Enforcement, Enforcement and Removal Operations. 2020d. COVID-19 Pandemic Response, Version 4. Available online: https://www.ice.gov/doclib/coronavirus/eroCOVID19responseReqsCleanFacilities-v4.pdf (accessed on 4 September 2020).

Immigration and Customs Enforcement, Enforcement and Removal Operations. 2020e. COVID-19 Pandemic Response Requirements, Version 5. Available online: https://www.ice.gov/doclib/coronavirus/eroCOVID19responseReqsCleanFacilities-v5.pdf (accessed on 27 October 2020).

Immigration and Customs Enforcement, Enforcement and Removal Operations. 2020f. Memorandum on Coronavirus Disease Action Plan, Revision 1. Available online: https://www.ice.gov/doclib/coronavirus/attF.pdf (accessed on 27 March 2020).

Immigration and Customs Enforcement, Enforcement and Removal Operations. 2021. COVID-19 Pandemic Response Requirements, Version 6. Available online: https://www.ice.gov/doclib/coronavirus/eroCOVID19responseReqsCleanFacilities.pdf (accessed on 16 March 2021).

Immigrations and Customs Enforcement, Family Residential Centers. 2020. Family Residential Standards (rev. 2020). Available online: https://www.ice.gov/doclib/frs/2020/2020family-residential-standards.pdf (accessed on 28 July 2021).

Immigration and Customs Enforcement, Immigration Health Services Corp. 2020. Interim Reference Sheet on 2019-Novel Coronavirus, Version 6. Available online: https://www.aila.org/infonet/ice-interim-reference-sheet-coronavirus (accessed on 6 March 2020).

Immigration and Customs Enforcement, Immigration Health Service Corps. 2021. Available online: https://www.ice.gov/detain/ice-health-service-corps (accessed on 24 March 2021).

Kalhan, Anil. 2010. Rethinking Immigration Detention, Columbia Law Review. *Sidebar*. pp. 42–58. Available online: http://www.columbialawreview.org/sidebar/volume/110/42_Anil_Kalhan.pdf (accessed on 25 October 2011).

Koulish, Robert. 2016. Immigration Detention in the Risk Classification Assessment Era. *Connecticut Public Interest Law Journal* 16: 1. Available online: https://cpilj.law.uconn.edu/wp-content/uploads/sites/2515/2018/10/16.1-Immigration-Detention-in-the-Risk-Classification-Assessment-Era-by-Robert-Koulish.pdf (accessed on 10 November 2016).

Melugin, Bill. 2021. FOX News, REPORT: ICE Confirms '30% of Detainees' Refusing COVID-19 Vaccine While in Detention Centers, 2021. Available online: https://candaceowensfans.com/report-ice-confirms-30-of-detainees-refusing-covid-19-vaccine-while-in-detention-centers/ (accessed on 22 July 2021).

Miner, Horace. 1956. Body Ritual among the Nacirema. *American Anthropologist* 58: 503–7. [CrossRef]

Morrissey, Kate. 2021. San Diego Union-Tribune, Court Ordered Return of Remain in Mexico Worsens Nightmare for Asylum Advocates. Available online: https://www.sandiegouniontribune.com/news/immigration/story/2021-08-28/court-remain-in-mexico-asylum (accessed on 28 August 2021).

Rosenberg, Mica, and Reade Levinson. 2018. Trump's Catch-and-Detain Policy Snares Many Who Have Long Called U.S. Home. Reuters. Available online: https://www.reuters.com/investigates/special-report/usa-immigration-court/ (accessed on 20 June 2018).

Schriro, Dora. 2009. U.S. Department of Homeland Security, Immigration and Customs Enforcement. Immigration Detention Overview and Recommendations. Available online: https://www.ice.gov/doclib/about/offices/odpp/pdf/ice-detention-rpt.pdf (accessed on 6 October 2009).

Smart, Noelle, Adam Garcia, and Nina Siulc. 2021. One Year Later, We Still Don't Know How Many People in ICE Detention Have Been Exposed to COVID-19, VERA Institute of Justice. Available online: https://www.vera.org/blog/one-year-later-we-still-dont-know-how-many-people-in-ice-detention-have-been-exposed-to-covid-19 (accessed on 8 April 2021).

U.S. Department of Homeland Security, Office of Inspector General. 2018. ICE's Inspections and Monitoring of Detention Facilities Do Not Lead to Sustained Compliance or Systemic Improvements. OIG-18-67. Available online: https://www.oig.dhs.gov/sites/default/files/assets/2018-06/OIG-18-67-Jun18.pdf (accessed on 26 June 2018).

U.S. Department of Homeland Security, Office of Inspector General. 2019. ICE Does Not Fully USE Contracting Tools to Hold Detention Facility Contractors Accountable for Failing to Meet Performance Standards. OIG-19-18. Available online: https://www.oig.dhs.gov/sites/default/files/assets/2019-02/OIG-19-18-Jan19.pdf (accessed on 29 January 2019).

U.S. Department of Homeland Security, Office of Inspector General. 2020. Early Experiences with COVID-19 at ICE Detention Facilities. OIG-20-42. Available online: https://www.oig.dhs.gov/sites/default/files/assets/2020-06/OIG-20-42-Jun20.pdf (accessed on 18 June 2020).

U.S. Department of Homeland Security, Office of Inspector General. 2021. ICE's Management of COVID-19 in Its Detention Facilities. Provides Lessons Learned for Future Pandemic Responses. OIG-21-58. Available online: https://www.oig.dhs.gov/sites/default/files/assets/2021-09/OIG-21-58-Sep21.pdf (accessed on 7 September 2021).

U.S. Immigration and Customs Enforcement Fiscal Year 2020 Enforcement and Removal Operations Report. 2021. Available online: https://www.ice.gov/doclib/news/library/reports/annual-report/eroReportFY2020.pdf (accessed on 12 May 2021).