# Peer review of "On the Other Side of the Looking Glass: COVID-19 Care in Immigration Detention"

_socsci, doi:10.3390/socsci10100353_

Round 1

Reviewer 1 Report

This paper is an extremely detailed and insightful exploration of the reasons why ICE has had a poor performance in protecting healthcare rights of immigration detainees in the framework of the coronavirus pandemic, with a special focus on legal and institutional factors. The paper is matchless in its capacity to closely monitor ICE’s structure, organisation and operations to map the causes of this institutional failure. Focusing exclusively on the US immigration detention system, the paper does not engage in a dialogue with the burgeoning literature on detention practices elsewhere. In addition, it does not put significant effort in relating its findings with the extant literature, which is largely absent from this piece. Both a certain comparative aspiration and the willingness to engage in a conversation with the scholarship on immigration detention might have further sharpened the considerable potential of this paper. However, as it currently stands, this contribution is highly valuable, since it provides a particularly well-informed, molecular perspective not only on what has happened in the field in immigration detention in the US since the onset of the pandemic, but also on the legal and institutional obstacles impeding detention facilities from providing adequate healthcare to detainees. Therefore, the paper should be published in the framework of this special issue. However, before bringing the paper to final decision, it might be improved by considering the following points:

  1. The main viewpoint to be missed in this paper is what may be called the ‘half full’ perspective. The paper persuasively stresses the shortcomings of the policies and practices adopted by ICE in managing the pandemic inside detention facilities. However, the paper seems to overlook certain additional aspects of ICE’s reaction to the coronavirus crisis that, at least seen from a distance, seem to be worthy of consideration. It has been pointed out that the DHS reached an all-time record in the number of detained noncitizens back in 2019, exceeding for the first time the symbolic threshold of half a million detainees. In stark contrast to this, the paper reports that the number of detainees in 2020 was a bit lower than 138,000 noncitizens (page 2). This suggests that the immigration enforcement apparatus was significantly downsized in 2020 (which is indirectly recognised in p. 5). If that is the case, the paper might want to more thoroughly analyse this significant change. Not in vain, although the US did not follow the particularly wide-ranging release policies implemented in a number of European jurisdictions in 2020, ICE nonetheless promoted a relatively generous release agenda that has had a significant impact on the immigration detention landscape in the US (see e.g. pp. 3, 18). The proposed analysis may be particularly pertinent in p. 5-6, where the paper highlights that the pandemic triggered a net-widening effect in ICE activities. This conclusion stands apparently in contrast to the detention data provided elsewhere in the paper (pp. 3, 5).
  2. The paper points out that ‘before the pandemic’ the number of people on ATD supervised by ICE was higher than that of detained noncitizens (p. 5). However, this consideration stands in contrast to the data provided in page 5 itself;
  3. The paper points out that in many aspects immigration detention facilities operate as prisons and actually give a prison treatment to detained noncitizens (p. 11). This point might remind readers of the widespread debate on the punitive nature of detention measures, and the implications of that punishment-like character. The paper apparently decries detention practices in the US for that reason. However, it does not elaborate why that penitentiary nature is so questionable. The reasons – at least, the legal reasons - why that is the case are particularly clear in the European scenario. Still, in the US, where detention facilities confine significant numbers of previously sentenced individuals, that criticism is not so self-evident – at least not for a non-specialised readership. The paper might want to further elaborate this point;
  4. The paper uses the word ‘incarcegration’ here and there (e.g. pp. 4, 6). Apparently, this is a notion that emphasises the carceral nature of detention practices. That being the case, it is similar – albeit apparently not identical - to the concept of ‘imm-carceration’ popularised by Ben Bowling. However, in contrast to Bowling’s notion, ‘incarcegration’ does not seem to be widely used by the extant literature. Consequently, the paper may want to define this notion;
  5. The paper incidentally uses acronyms which are widely used in the US but hard to understand elsewhere. The vast majority of these acronyms are defined, but some of them are not, or they are not defined when they are first used (e.g. FY, ERO). The paper may want to define these acronyms, so as to allow international scholars to easily follow its narrative;
  6. An additional editing effort is needed. The paper still contains a certain number of typos (e.g. it is described as ‘chapter’ in page 3). Some of these typos are relatively significant, e.g. the paper points out that ‘58%’ of the individuals in ICE’s custody in June 2021 had no criminal convictions (page 4), whilst some lines below it mentions that this percentage is actually ‘28%’ (p. 5). In addition, page 15 contains two paragraphs that are identical;

Needless to say, all these comments do not challenge the particularly positive assessment of this paper, which unequivocally deserves to be published. By taking into consideration the aforementioned points, though, the paper might further improve its already valuable contribution to the conversation on immigration detention and the covid pandemic.

Reviewer 2 Report

I have no doubts that this paper should be published as it brings new set of very interesting and really up to date information about immigrant detention in the US. But it should be carefully redacted. At the moment the paper is filled with a huge number of details what makes it very difficult to read and to follow the main argument presented by the Author.

Some parts should be significantly shortened – like the ones in the first part of the paper which gave an overview of the detention in the US. That refers especially to chapters “The Incarcergration of Immigration Detention” and “Government Oversight”. The very good example in this regards is subchapter “ICE’s Oversight” on page 9 which is mostly off-topic of this paper and reference to the Covid-19 pandemic are marginal. It could be removed without any harm to the paper. The same refers to chapter “ICE Risk Assessment”.

The paper also lacks in academic references and theoretical background. It looks more like a very good report from the field research and less like a proper academic paper. Adding by the Author some academic backdrop and analysing detention practices during Covid-19 against it would be very valuable and should strong the paper significantly. A number of theories could serve this purpose, like – to name just a few – referring to state violence (or state crimes), proliferation of punishment and pain (and preventive justice which covers immigrant detention practices) etc.

Some statements by the Author need to be supported by proper references – to prove that they are not a subjective opinions of the Author themself but they present facts that the Authors relies on – like in the second paragraph on p. 6 about alternatives to detention (another question is though, is information on ATD crucial to this paper and should they be kept in it or not) or the second paragraph on p. 11 on private prisons.

Some minor issues:

  • Table 2 fully repeat information from table 3 (but it’s less developed) and should be removed;
  • Last paragraph on p. 15 is again a repetition of the former paragraph;
  • The word ‘Incarcergration’ should be replaced by ‘Incarceration’ – this misspelling appears several times throughout the paper.
